# Efficient RL Training for Reasoning Models via Length-Aware Optimization

## Abstract

Long reasoning models have demonstrated remarkable performance on reasoning tasks but often incur a long reasoning path with significant memory and time costs. Existing methods primarily aim to shorten reasoning paths by introducing additional training data and stages. In this paper, we propose three critical reward designs integrated directly into the rule-based reinforcement learning process of long reasoning models, which reduce the response length without extra training stages. Experiments on four settings show that our method significantly decreases response length while maintaining or even improving performance. Specifically, in a logic reasoning setting, we achieve a 40% reduction in response length averaged by steps alongside a 14% gain in performance. For math problems, we reduce response length averaged by steps by 33% while preserving performance.

## 1 Introduction

Recent advancements in long reasoning models (LRMs) have demonstrated exceptional performance across diverse reasoning tasks. Leveraging large-scale, rule-based reinforcement learning (RL), these models have developed advanced cognitive capabilities, including self-reflection, self-critique, and self-correction Chen et al. (2025a); DeepSeek-AI et al. (2025); OpenAI et al. (2024) A defining feature of reasoning models is the progressive increase of reasoning length during training, which often correlates with improved reasoning abilities DeepSeek-AI et al. (2025). Longer reasoning length enables models to explore intricate solution paths, decompose complex problems, and arrive at more accurate conclusions.

However, increased reasoning length introduces significant challenges. During inference, longer responses lead to higher computational costs and heavier KV caches, drastically slowing down the decoding process. During training, the growing response length considerably slows down the training process, and may even make large-scale training on specific tasks impractical DeepSeek-AI et al. (2025). Despite the advantages of longer reasoning paths, recent studies have shown that longer reasoning paths do not necessarily lead to better performance Fatemi et al. (2025); Chen et al. (2025b); Team et al. (2025); Yang et al. (2025). In some cases, overly long reasoning paths can lead to inefficiencies or even degraded performance, as models may overthink or generate redundant steps Chen et al. (2025b); Sui et al. (2025).

Existing methods for reducing redundant response length in LRMs have primarily relied on supervised fine-tuning or off-policy RL strategies Xia et al. (2025); Kang et al. (2024); Ma et al. (2025b); Munkhbat et al. (2025); Yu et al. (2024); Liu et al. (2024); Cui et al. (2025); Luo et al. (2025a); Shen et al. (2025). However, these approaches are not directly applicable to the on-policy RL frameworks commonly used in LRMs training. One promising approach, the direct length-reward method proposed by Kimi Team et al. (2025), incorporates response length as a factor in the RL reward function. While this method shows potential, our reproduction of Kimi's length reward reveals significant limitations. When applied early in the RL training process, it drastically shortens response length but disrupts the model's exploratory behavior, leading to suboptimal performance. Moreover, other works Arora & Zanette (2025); Hou et al. (2025) also show degraded performance. This highlights the need for a approach that can be directly applied in the on-policy RL training.

To address this challenge, we propose a novel method, Short-RL, designed to regulate response length during RL training without compromising model performance. Through a detailed analysis of the Kimi length-reward approach, we identify its adverse effects on learning dynamics, particularly its

tendency to suppress reasoning diversity in the early stages of training. Motivated by these findings, we introduce three innovative enhancements to the length-reward framework, each aimed at balancing efficiency and reasoning quality:

- **Correctness-Conditioned Length Reward**: reward computation is restricted to correctly answered samples. This design aims to minimize the impact of length penalties on the exploration behaviors of model reasoning.
- **Neutral Length Zone**: exempts responses within an acceptable length range from length penalties, allowing the model to retain flexibility in exploring responses with appropriate lengths.
- **Accuracy-Aware Length Reward**: automatically disables length rewards when batch accuracy falls below a specified threshold.

Our approach effectively regulates response length during training without compromising—and in some cases enhancing—model performance. Experimental results on logical reasoning tasks show a 40% average reduction in response length during training, alongside a 14% improvement in evaluation scores. In the mathematical reasoning setting, our method achieves a 33% reduction in average response length while maintaining performance comparable to standard RL training.

## 2 RELATED WORK

### 2.1 REASONING MODELS TRAINED WITH RULE-BASED RL

Large reasoning models are renowned for their exceptional performance across various reasoning tasks. By engaging in extensive deliberation before generating a final answer, these models exhibit human-like complex reasoning capabilities DeepSeek-AI et al. (2025); OpenAI et al. (2024). Notably, DeepSeek-AI et al. (2025) demonstrated that large-scale, rule-based reinforcement learning (RL) can significantly enhance the reasoning abilities of large language models (LLMs). However, the growing response length during training introduces substantial memory and computational overhead, hindering both training and inference efficiency—and in some cases, even rendering large-scale RL infeasible for specific tasks DeepSeek-AI et al. (2025).

Existing efforts to replicate the RL process of DeepSeek-R1 have primarily focused on domain-specific datasets. For instance, Xie et al. (2025) achieved promising results on a logic puzzle dataset Xie et al. (2024), while other works have explored rule-based RL training in mathematical domains Zeng et al. (2025); Luo et al. (2025b); Hu et al. (2025); Yu et al. (2025). These studies observe a trend of increasing response lengths during training, further underscoring the need for efficient optimization methods.

### 2.2 LONG TO SHORT LLM REASONING

The lengthy reasoning processes of language models incur significant memory and time costs, prompting numerous approaches to reduce the reasoning length.

Existing methods for shortening responses primarily operate in either supervised fine-tuning settings Xia et al. (2025); Kang et al. (2024); Ma et al. (2025b); Munkhbat et al. (2025); Yu et al. (2024); Liu et al. (2024); Cui et al. (2025) or off-policy reinforcement learning frameworks Luo et al. (2025a); Shen et al. (2025). However, these techniques demand additional training stages and curated datasets, making them incompatible with the in-process reinforcement learning of long-reasoning models. Furthermore, their efficacy on post-trained models remains unverified. There is also active research on prompt-guided efficient reasoning, which seeks to reduce response length through prompt engineering Han et al. (2025); Renze & Guven (2024); Xu et al. (2025); Ma et al. (2025a). While promising, these methods tend to be task-specific and often degrade overall model performance.

Other lines of work investigate shortening reasoning through model merging or collaborative agent frameworks She et al. (2025); Wu et al. (2025). Additionally, some approaches propose dynamically routing reasoning behavior based on the input question or user intent Anthropic (2025); Aytes et al. (2025); Chuang et al. (2025); Ong et al. (2025); Pu et al. (2025); Aggarwal & Welleck (2025).

Direct length-based rewards for on-policy RL, first proposed by Kimi 1.5 Team et al. (2025), are restricted to post-RL applications. As noted by Kimi, applying such rewards during initial training

impedes training convergence—a finding corroborated by our experiments, which reveal further limitations of this approach. Arora & Zanette (2025) proposes to scale the correct answer reward based on response length. Nonetheless, their approach reveals a trade-off between response length and model performance. Additionally, their experiments are confined to only around 100 training steps, leaving the long-term implications unexplored. Hou et al. (2025) proposes penalizing correct responses that exceed a length limit, though their method highlights an inherent trade-off between response brevity and model accuracy. In this work, we will demonstrate the severe shortcomings of such length-based rewards under extended training regimes.

## 3 METHODOLOGY

### 3.1 LENGTH-AWARE OPTIMIZATION

A straightforward approach to reducing reasoning length is to incorporate a length penalty into the original reward function. Generally, the length reward can be incorporated into the rule-based reward as follows:

$$R(x, y) = C(y) + \alpha \cdot S(y), \qquad (1)$$

where $C(y)$ denotes the rule-based reward and $S(y)$ denotes the length reward. $\alpha$ is a coefficient.

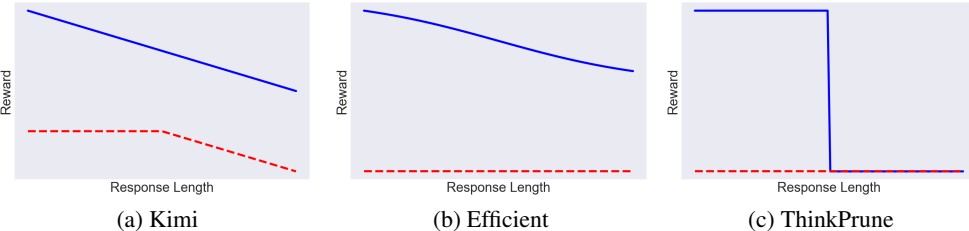

| (a) Kimi | (b) Efficient | (c) ThinkPrune |

Figure 1: Reward values as a function of response length, where blue lines indicate rewards for correct responses and red lines represent rewards for incorrect responses.

Kimi Team et al. (2025) initially proposes their length reward function. However, Kimi's length reward mechanism cannot be directly applied during the early stages of the reinforcement learning training process. Instead, the reward is only introduced during a post-RL training phase.

Subsequently, two other length-based reward functions were proposed (Efficient Arora & Zanette (2025); ThinkPrune Hou et al. (2025)). While these approaches differ in their usage settings, they exhibit similar limitations that can hinder the performance of RL training. A brief visualization of those rewards (combined with rule-based rewards) is plotted in Figure 1.

In this work, we primarily focus on the Kimi length reward, though the reward design we propose is broadly applicable to other length-based reward functions as well.

#### 3.1.1 LENGTH REWARD IN KIMI

Suppose a response is defined by $(y_i, z_i)$, where $y_i$ represents the answer and $z_i$ the reasoning process. Given a set of sampled responses $(y_1, z_1), \ldots, (y_k, z_k)$ for a problem $x$ with the correct answer $y^*$, let $\ell_i$ denote the length of response $(y_i, z_i)$. Define $\ell_{\min} = \min_i \ell_i$ and $\ell_{\max} = \max_i \ell_i$. If $\ell_{\max} = \ell_{\min}$, the length reward is set to zero for all responses. Otherwise, the length reward is defined as:

$$\text{reward}_{\text{len}}(i) = \begin{cases} \lambda & \text{if } r(x, y_i, y^*) = 1 \\ \min(0, \lambda) & \text{if } r(x, y_i, y^*) = 0 \end{cases}, \quad \text{where } \lambda = 0.5 - \frac{\ell_i - \ell_{\min}}{\ell_{\max} - \ell_{\min}}. \qquad (2)$$

Kimi introduces a weighted adjustment to this reward by scaling it with a factor $\alpha$ before adding it to the rule-based reward.

#### 3.1.2 LIMITATIONS OF LENGTH REWARD IN EARLY TRAINING

In the original Kimi 1.5 paper Team et al. (2025), the length reward is not applied during the initial stage of reinforcement learning training. Instead, standard policy optimization is performed first,

and a constant length penalty is introduced only in the later training phase. The authors claim that applying the length reward too early negatively affects training stability and convergence.

To investigate this claim, we reproduced the experimental setup of Logic-RL Xie et al. (2025) and modified it to include the Kimi length reward from the beginning of training. Specifically, we varied the weight coefficient $\alpha$ across the values $[1, 0.5, 0.1, 0.01]$, keeping all other hyperparameters fixed. We then evaluated the resulting models on the `ppl5` dataset (logic puzzles with 5 people) Xie et al. (2024), measuring both test accuracy and average response length. All models were trained from scratch for 3 epochs. As shown in Figure 2, directly incorporating the length reward from the start results in a reward hacking phenomenon, with response lengths rapidly collapsing to very short outputs.

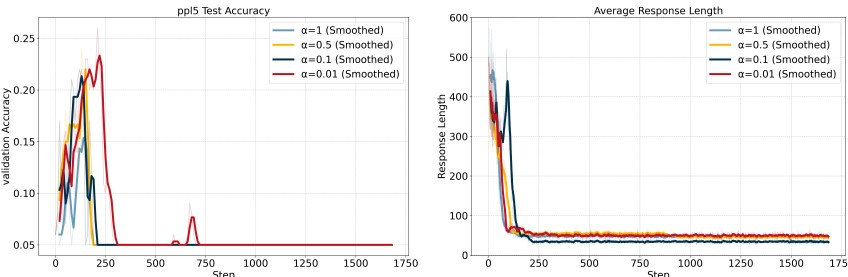

Figure 2: Test accuracy (left) and average response length (right) across different values of $\alpha$.

## 3.2 SHORT-RL

In this subsection, we identify two major issues with the direct length reward proposed by Kimi and introduce three key reward design principles that are critical for optimizing model performance. The first two focus on preserving model exploration and output diversity, while the third is aimed at maintaining overall task performance.

### 3.2.1 PROBLEM 1: LENGTH REWARD BIAS AS A BARRIER TO EXPLORATORY BEHAVIOR

The $\ell_{\min}$ and $\ell_{\max}$ values defined by Kimi are computed based on all responses to a given problem $x$. Furthermore, Kimi applies the length reward function $\text{reward}_{\text{len}} = \min(0, \lambda)$ when the answer is incorrect. This leads to longer incorrect responses being penalized more severely than shorter ones. Additionally, the reward function is formulated as a linear function that favors shorter responses, assigning them higher rewards while penalizing longer ones. This design incentivizes convergence toward the shortest possible outputs, thereby diminishing response diversity.

These two aspects of the reward function suppress model exploration and increase the risk of the model converging to suboptimal local minima. Notably, a similar limitation is observed in the reward formulation proposed by Arora & Zanette (2025).

To evaluate this effect, we track the diversity metric associated with the Kimi length reward ($\alpha = 0.1$) during training over the course of one epoch. The diversity metric is computed as the average of semantic diversity Guo et al. (2024a), lexical diversity (measured using the distinct-n metric Li et al. (2016); Guo et al. (2024a)) and syntactic diversity (measured using a graph-based metric Guo et al. (2024b;a)). We track the diversity metric on `ppl5` test dataset each 50 steps. As illustrated in Figure 3 (blue line), the Kimi length reward leads to a gradual reduction in output diversity.

To address this issue, we propose two reward design modifications that help preserve model diversity:

**Reward Design I: Correctness-Conditioned Length Reward**

We propose that length-based rewards should be applied only to correct responses. Specifically, the length reward is computed exclusively for correct answers, with $\ell_{\min}$ and $\ell_{\max}$ calculated solely from correct responses to each question. This approach is similar to the reward scaling strategy adopted by Arora & Zanette (2025) and Hou et al. (2025), who similarly restrict reward adjustments to correct outputs.

**Reward Design II: Neutral Length Zone**

To avoid penalizing responses that fall within an acceptable length range, we introduce a hyperparameter $\tau_\ell$, referred to as the *length tolerance*. For correct responses, the length reward is defined as follows:

- If the response length $\ell(i)$ satisfies $\ell(i) \leq \ell_{\min} + \tau_\ell$, the length reward is set to 0.5, matching the reward for the shortest correct response.
- For responses exceeding this threshold, the length reward is set to the value $\lambda$ as defined earlier.

We evaluate the effectiveness of our reward modifications—using $\tau_\ell = 200$ and $\alpha = 1$—during training. As illustrated in Figure 3, the Kimi length reward reduces model output diversity, while design I and design II (red line) successfully preserve it.

### 3.2.2 Problem 2: Instability Performance due to Length Penalty

In the Kimi setting, the length reward is applied at every training step, regardless of model performance or prediction quality. That is, each gradient update includes a penalty on longer responses. In our experiments, we find that although Design I and Design II help retain response diversity, in some cases, model performance is still degraded. As shown in Figure 4b, the red curve (D1+D2), corresponding to our proposed Design I and Design II combination, exhibits unstable performance across training. We hypothesize that this inconsistency arises from the complex and dynamic relationship between output ac-

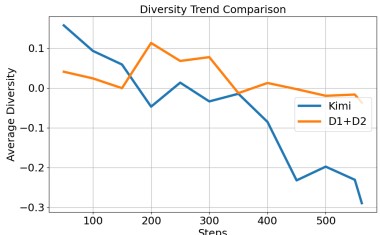

Figure 3: The diversity metric on `ppl5` test dataset.

curacy and reasoning length. Specifically, while Designs I and II encourage concise outputs, the model may, at particular training stages, require extended reasoning paths to arrive at correct answers and to develop new reasoning capabilities. In such cases, penalizing longer outputs too aggressively may hinder the learning process.

To address this issue, we propose to stop the application of the length reward until the training process has stabilized—namely, when batch accuracy shows consistent improvement. This ensures that the model first learns to produce correct and robust outputs before being incentivized to optimize for brevity.

**Reward Design III: Accuracy-Aware Length Reward**

We define a hyperparameter $\tau_{\text{acc}}$ that controls the accuracy threshold. For each training batch, we compute the batch accuracy acc over all rollout samples, and maintain $\text{acc}_{\max}$, the maximum accuracy achieved up to that point in training. The length reward is applied only when the condition $\text{acc} \geq \text{acc}_{\max} - \tau_{\text{acc}}$ is satisfied.

### 3.3 Short-RL Length Reward

Based on the above analysis and our proposed three reward designs, we integrate their key insights into a unified reward formulation. Specifically, we combine the advantages of conditional reward application (Design III), adaptive reward strength based on output length (Design II), and correctness filtering (Design I) to construct a robust length reward function. This design ensures that rewards are only provided when predictions are correct, model accuracy is stable, and output length deviates meaningfully from the minimal correct length. The final length reward function is defined as:

$$\text{reward}_{\text{len}}(i) = \begin{cases} \beta, & \text{if } r(x, y_i, y^*) > 0 \text{ and acc} \geq \text{acc}_{\max} - \tau_{\text{acc}} \\ 0, & \text{otherwise} \end{cases},$$

$$\text{where} \quad \beta = \begin{cases} \lambda, & \text{if } \ell(i) > \ell_{\min} + \tau_\ell \\ 0.5, & \text{otherwise} \end{cases}, \tag{3}$$

$$\lambda = 0.5 - \frac{\ell(i) - \ell_{\min}}{\ell_{\max} - \ell_{\min}},$$

$$\ell_{\min} = \min_j \ell(j), \quad \ell_{\max} = \max_j \ell(j), \quad \text{where } j \in \{j \mid y_j = y^*\}.$$

Here, $\tau_{\mathrm{acc}}$ controls the sparsity of the reward: a smaller value results in a sparser reward signal. When $\tau_{\mathrm{acc}} = 1$, the scheme reduces to a dense reward. Meanwhile, $\tau_\ell$ determines the allowed deviation from the minimum correct response length $\ell_{\min}$; when $\tau_\ell = 0$, the function simplifies to the linear reward used in Kimi.

# 4 EXPERIMENTS

## 4.1 EXPERIMENTAL SETTINGS

We evaluate our method across two distinct domains: logic reasoning and mathematical reasoning. The logic reasoning domain is represented by the Logic-RL project Xie et al. (2025), while the mathematical reasoning domain includes three settings: DeepScaleR Luo et al. (2025b), SimpleRL-Reason Zeng et al. (2025), and Open-Reasoner-Zero Hu et al. (2025). In all experiments, we employ the same model architecture and training framework Sheng et al. (2024) as used in the original projects. For the three mathematical reasoning settings, we use a prompt template similar to DeepSeek-R1, and a format reward is also included in the standard reward. Details can be found in Training Details.

### 4.1.1 LOGIC REASONING

We use the same dataset as Logic-RL and initialize the model from the Qwen2.5-7B base model Qwen et al. (2025). The hyperparameters for Short-RL are set as $\tau_\ell = 200$, $\tau_{\mathrm{acc}} = 0.05$, and $\alpha = 1$. Additional implementation details are provided in Appendix Training Details.

We evaluate the final accuracy on 2- to 8-person tasks using Logic-RL's evaluation script.

To assess generalization, we also evaluate out-of-domain performance on the AIME and AMC benchmarks following Logic-RL's protocol. Additionally, we report two token-length metrics: (1) step-wise average response length during training, reflecting training speed, and (2) average response length at the final step, indicating inference speed after training.

### 4.1.2 MATH REASONING

We conduct comparative experiments on three settings. Nearly all hyperparameters are retained from the original implementation. For Short-RL, the hyperparameters settings can be found in Table 3 of Appendix Training Details. Further implementation details are available in Appendix Training Details.

Evaluation is carried out across five benchmark datasets: AIME2024invitational mathematics examination (2024), AMC23AI-MO (2025), MATH-500Lightman et al. (2023), Minerva MathHendrycks et al. (2021), and Olympiad BenchHe et al. (2024). We also report two token-length metrics: (1) the step-wise average response length during training, and (2) the average token length at the final step.

### 4.1.3 BASELINES

We compare our method with the following baselines:

- **Standard**: Reinforcement learning with standard rule-based rewards.

- **Kimi**: Rule-based rewards augmented with the Kimi length reward ($\alpha = 1$). **Note that the Kimi length reward was originally applied in a post-RL stage after a standard RL stage. Directly applying this reward function may lead to issues and varying the choice of $\alpha$ remains susceptible to reward hacking (discussed in Section 3.1.2)**. Thus we provide a Kimi (post) baseline to show the best performance of Kimi reward function applied after the standard RL. For this two-stage approach, we report the step-wise average response length during the first (standard RL) stage in the tables.

- **Efficient**: A length-aware scaling reward from (Arora & Zanette, 2025), where we select optimal $\alpha$ values from $0.02, 0.05, 0.08, 0.10$ for each method: Logic-RL ($\alpha = 0.05$), DeepScaleR ($\alpha = 0.10$), and both SimpleRL-Reason and Open-Reasoner-Zero ($\alpha = 0.02$). Note that the $\alpha$ used in

Table 1: Logic-RL valuation on the final checkpoint.

| Method | In Domain | | | | | | | | Out of Domain | | Average Response Length | |
|--------|-----------|---|---|---|---|---|---|---------|-----|------|-------------------------|------|
| | ppl2 | ppl3 | ppl4 | ppl5 | ppl6 | ppl7 | ppl8 | Average | AMC | AIME | Averaged by Steps | Last |
| Standard | 82 | 87 | 88 | 81 | 76 | 69 | 70 | 79 | 39.76 | 7.77 | 1477 | 2632 |
| Kimi (post) | 84 | 88 | 89 | 84 | 79 | 74 | 76 | 82 | 39.89 | 8.13 | 1477 | 763 |
| Efficient | 76 | 81 | 79 | 77 | 62 | 48 | 51 | 68 | 37.35 | 7.77 | 772 | 843 |
| ThinkPrune | 80 | 84 | 86 | 82 | 70 | 66 | 64 | 76 | 38.47 | 7.35 | 832 | 793 |
| Short-RL | **97** | **97** | **99** | **95** | **92** | **83** | **87** | **93** | **42.17** | **8.74** | 889 | 535 |

Table 2: Evaluation of math reasoning.

| Model | Math Benchmarks | | | | | | Average Response Length | |
|-------|-----------------|-------|---------|--------------|----------------|---------|-------------------------|------|
| | AIME2024 | AMC23 | MATH500 | Minerva Math | Olympiad Bench | Average | Averaged by Steps | Last |
| *DeepScaler* | | | | | | | | |
| Standard | 26.67 | 59.04 | **81.40** | **26.10** | 42.65 | 47.17 | 2523 | 3072 |
| Kimi (post) | 23.33 | **61.45** | 81.00 | 25.37 | **42.79** | 46.79 | 2523 | 1678 |
| Efficient | 20.00 | 49.40 | 57.8 | 16.54 | 33.73 | 35.49 | 1517 | 1537 |
| ThinkPrune | 26.67 | 56.63 | 78.40 | 25.74 | 41.31 | 45.75 | 1589 | 1621 |
| Short-RL | **30.00** | **60.24** | 80.60 | **26.47** | 42.65 | **47.99** | 1692 | 1700 |
| *Open Reasoner Zero* | | | | | | | | |
| Standard | 16.67 | **50.60** | **78.80** | 30.88 | 38.04 | 43.00 | 746 | 840 |
| Kimi (post) | **20.00** | 49.40 | 77.40 | **31.25** | **38.63** | **43.34** | 746 | 621 |
| Efficient | 13.33 | 46.99 | 66.40 | 26.47 | 35.96 | 37.83 | 578 | 655 |
| ThinkPrune | 13.33 | 48.19 | 76.80 | 27.57 | 37.15 | 40.61 | 677 | 682 |
| Short-RL | 16.67 | **50.60** | 78.60 | 30.52 | 38.19 | 42.92 | 660 | 670 |
| *SimpleRL-Reason* | | | | | | | | |
| Standard | 13.33 | 48.19 | 77.00 | **32.72** | **39.97** | 42.24 | 703 | 791 |
| Kimi (post) | 16.67 | 48.19 | 77.40 | 31.99 | 39.67 | 42.78 | 703 | 601 |
| Efficient | 6.67 | 38.55 | 64.8 | 22.06 | 28.68 | 32.15 | 492 | 532 |
| ThinkPrune | 10.00 | 46.99 | 69.40 | 31.62 | 37.30 | 39.06 | 613 | 598 |
| Short-RL | **20.00** | **49.40** | **78.20** | **32.72** | 39.23 | **43.91** | 554 | 620 |

Efficient (as a scaling factor) differs from the $\alpha$ used in our method. Additionally, the experimental results in their paper already show an obvious trade-off between accuracy and response length.

- **ThinkPrune**: A length-aware cosine reward proposed by (Hou et al., 2025). We select the length limit that yields a comparable average response length to our method: 1700 for Logic-RL, 2500 for DeepScaler, 1500 for OpenReasonerZero and SimpleRL-Reason. Similarly, the experimental results in their paper show a performance trade-off.

## 4.2 MAIN RESULTS

### 4.2.1 LOGIC REASONING

As is shown in Table 1, our proposed Short-RL method effectively regulates response length while consistently outperforming standard RL approaches in terms of accuracy. Specifically, Short-RL achieves a 40% reduction in step-wise average response length while delivering statistically significant accuracy gains across all evaluated tasks. In contrast, the Efficient and ThinkPrune baselines exhibit inferior performance. Although Kimi (post) eventually achieves strong accuracy and inference efficiency, it underperforms in training efficiency, as reflected by its higher step-wise average response length.

### 4.2.2 MATH REASONING

Quantitative evaluation in Table 2 reveals that Short-RL achieves 33% , 11% , 21% reduction in step-averaged response length compared to standard RL approaches across the three settings respectively. In contrast, Efficient and ThinkPrune baselines demonstrate poorer performance. Kimi also underperforms in training efficiency despite achieving similar accuracy.

## 5 ABLATION STUDY

### 5.1 COMPONENT ABLATION

We conduct an ablation study to evaluate the impact of our proposed designs. All experiments are performed on the Logic-RL dataset and use the same length reward weight $\alpha = 1$.

We compare the accuracy and average response length curves across several configurations. D1 applies standard RL using our proposed length reward design I. D1+D2 incorporates both design I and design II, while D1+D3 combines design I and design III. Finally, our proposed method, Short-RL, integrates all three designs: I, II, and III.

As shown in Figure 4, our proposed reward designs significantly improve both response length control and validation accuracy. The subfigure 4a shows that the Standard baseline generates overly long responses, while the Kimi baseline collapses to very short outputs. In contrast, our designs (D1, D1+D2, D1+D3) progressively stabilize length generation, with Short-RL achieving the most balanced outcome. The subfigure 4a illustrates consistent accuracy gains from our designs. Short-RL consistently achieving the highest accuracy. Even partial configurations (D1+D2, D1+D3) outperform Kimi, underscoring the effectiveness and complementarity of each design component. Note that the ppl5 test set here differs from the final evaluation set, following the practice of Logic-RL.

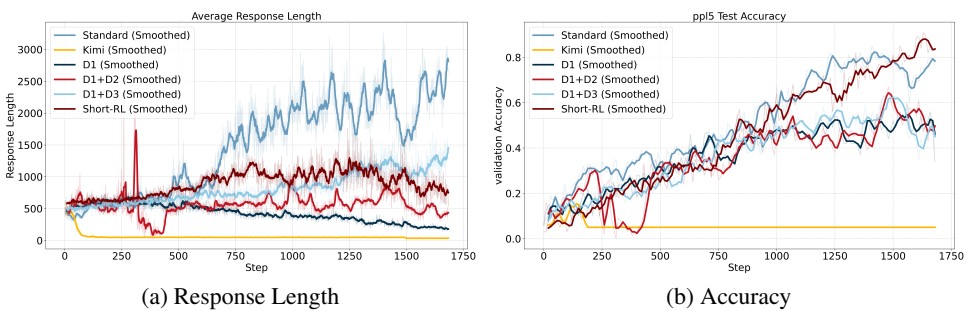

(a) Response Length      (b) Accuracy

Figure 4: Ablation study on three reward designs

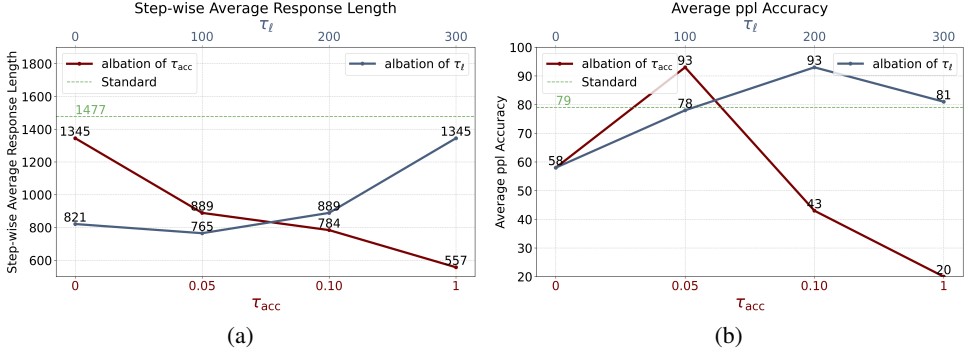

(a)      (b)

Figure 5: Ablation study on the impact of length tolerance and accuracy tolerance, with both factors plotted on a shared y-axis. The upper x-axis represents the length tolerance, while the lower x-axis represents the accuracy tolerance.

### 5.2 IMPACT OF LENGTH AND ACCURACY TOLERANCE

We vary the $\tau_\ell$ among 0, 100, 200, and 300, while fixing the $\tau_{acc}$ to 0.05. The comparisons of step-wise average response length and average accuracy on ppl tasks are shown in Figure 5. We observe that an overly small length tolerance (e.g., 0) leads to shorter average responses and degraded

performance. But the model is not too sensitive to the choice of length tolerance. Varying the choice among 100, 200, 300 still achieves good performance. Larger length tolerance may result in longer average response length. For this setting, a length tolerance of around 200 achieves the best balance.

We vary $\tau_{\mathrm{acc}}$ among 0, 0.05, 0.10, and 1.0, while fixing the $\tau_\ell$ to 200. Figure 5 shows that model performance is sensitive to this parameter. Specifically, higher $\tau_{\mathrm{acc}}$ (e.g., 1.0) leads to shorter response lengths and degraded performance. A good choice in this setting may be around 0.05.

# 6 TRACK THE LENGTH REWARD

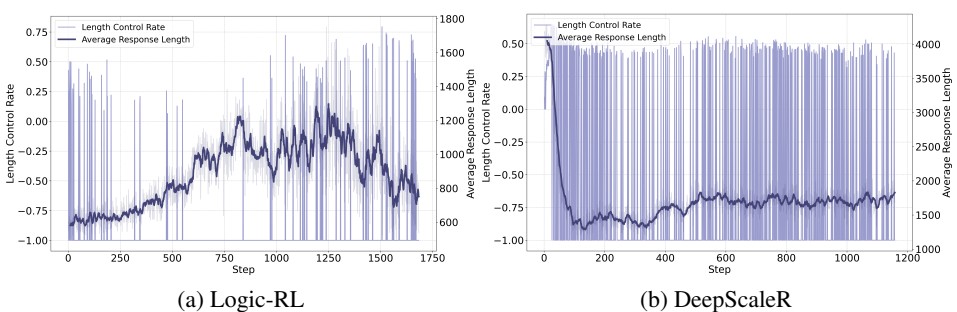

(a) Logic-RL  (b) DeepScaleR

Figure 6: Tracking the length reward during training

During training, we monitor the application of length rewards. We introduce a batch-wise metric called length control rate ($\gamma_\ell$). For each batch, let $N$ be the number of correct responses. Among these, $R$ denotes the number of responses with $\mathrm{reward}_{\mathrm{len}} < 0.5$. We then define:

$$\gamma_\ell = \begin{cases} \frac{R}{N}, & \text{if } N \neq 0 \text{ and acc} \geq \mathrm{acc}_{\max} - \tau_{\mathrm{acc}} \\ 0, & \text{if } N = 0 \\ -1, & \text{if acc} < \mathrm{acc}_{\max} - \tau_{\mathrm{acc}} \end{cases}, \tag{4}$$

We track the proposed metrics and the average response length during training in two experiments, as shown in Figure 6. We observe that the length reward is distributed throughout the training process. In DeepScaleR, length rewards are applied more frequently. The curves for SimpleRL-Reason and Open-Reasoner-Zero can be found in Appendix Track the Length Reward.

# 7 LIMITATIONS

Our method is designed for tasks where responses consist of a reasoning process followed by a short definitive answer (e.g., math, logic). In such settings, lengthy reasoning often contains redundant steps, making efficiency improvements viable. However, for tasks like creative writing, where reasoning is minimal or stylistic variation is valuable, favoring shorter outputs may not be appropriate.

Moreover, reward designs II and III rely on manual hyperparameter tuning, which may require adaptation across different tasks or models.

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

# A  APPENDIX

## A.1  ADDITIONAL EXPERIMENTS

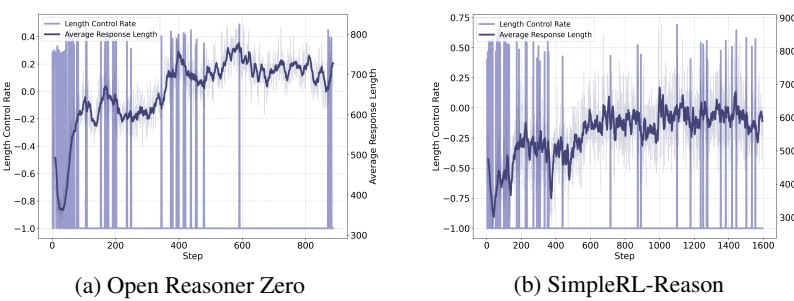

(a) Open Reasoner Zero    (b) SimpleRL-Reason

Figure 7: Visualization of the length control rate during train- ing.

| Setting | Logic-RL | DeepScaleR | Open Reasoner Zero | SimpleRL-Reason |
|---|---|---|---|---|
| learning rate | 1e-6 | 1e-6 | 5e-7 | 5e-7 |
| batch size | 8 | 128 | 64 | 16 |
| ppo_mini_batch_size | 32 | 64 | 256 | 64 |
| ppo_micro_batch_size | 8 | 32 | 64 | 2 |
| rollout_n | 8 | 8 | 8 | 8 |
| temperature | 0.7 | 0.6 | 1.0 | 1.0 |
| kl_loss_coef | 0.001 | 0.001 | 0.001 | 0.0001 |
| epochs | 3 | 3 | 1 | 3 |
| max_response_length | 4096 | 8192 | 4096 | 8192 |
| algorithm | reinforce++ | grpo | grpo | grpo |
| $\tau_\ell$ | 200 | 100 | 100 | 50 |
| $\tau_{acc}$ | 0.05 | 0.05 | 0.02 | 0.05 |
| $\alpha$ | 1 | 1 | 1 | 1 |
| Model | Qwen2.5-7B | DeepSeek Distill Qwen-1.5B | Qwen2.5-7B | Qwen2.5-7B |

Table 3: Training details.

### A.1.1 TRACK THE LENGTH REWARD

We also track the metric defined in Section Track the Length Reward in Figure 7 (Open Reasoner Zero and SimpleRL-Reason).

```
<|im_start|>system\nYou are a helpful assistant. The
assistant first thinks about the reasoning process in
the mind and then provides the user with the answer. The
reasoning process and answer are enclosed within <think>
</think> and<answer> </answer> tags, respectively, i.e.,
<think> reasoning process here </think><answer> answer
here </answer>.  Now the user asks you to solve a
logical reasoning problem. After thinking, when you
finally reach a conclusion, clearly state the identity
of each character within <answer> </answer> tags. i.e.,
<answer> (1) Zoey is a knight\n(2) ...
</answer>.\n<|im_end|>\n<|im_start|>user\n{quiz}\n<|im_e
nd|>\n<|im_start|>assistant\n<think>
```

```
The user asks a question, and the Assistant solves
it.The assistant first thinks about the reasoning
process in the mind and then provides the user with the
final answer. The reasoning process and answer are
enclosed within <think> </think> and <answer> </answer>
tags, respectively, i.e., <think> reasoning process here
</think><answer> answer here </answer>.
\n\nUser:{question}\nAssistant: <think>
```

(a)                                                      (b)

Figure 8: The prompt template for Logic-RL and Math-RL.

### A.2 TRAINING DETAILS

Our experiments were conducted using a compute node equipped with 8 NVIDIA H100 GPUs. The CUDA version we use is 12.3.

### A.2.1 LOGIC-RL TRAINING AND EVALUATION DETAILS

The training and evaluation prompt template (Figure 8a) used in Logic-RL remains the same as in the original GitHub project. The training hyperparameters are listed in Table 3. During evaluation, we directly use the code from Logic-RL, which applies a temperature of 1.0 and top_p=1.0 for logic tasks, and a temperature of 0.8 with top_p= 0.95 for math tasks.

### A.2.2 TRAINING AND EVALUATION DETAILS FOR MATH

The training and evaluation prompt template for three math settings is shown in Figure 8b. The training hyperparameters are listed in Table 3. During evaluation, we directly use the code from DeepScaleR, which employs a temperature of 1.0.

### A.2.3 REWARD DETAILS

In all the math experiments, the standard reward employs a format and outcome-based reward scheme. That is:

$$\text{reward} = \begin{cases} 3 & \text{, the format is correct and the answer is right} \\ -0.5 & \text{, the format is correct and the answer is wrong} \\ -3 & \text{, the format is wrong} \end{cases} . \quad (5)$$

In Logic-RL experiments, we directly use their original standard reward design.

