# OpenReview forum: "Efficient RL Training for Reasoning Models via Length-Aware Optimization"
_ICLR.cc/2026/Conference — ICLR 2026 Conference Withdrawn Submission_

### Official Review · Reviewer_MHBf · 2025-10-25

**Soundness:** 3
**Presentation:** 2
**Contribution:** 2
**Rating:** 2
**Confidence:** 4

**Summary:**

The authors propose Short RL, a length aware reward shaping method with three key designs: applying length reward only on correct samples, setting a neutral length range, and toggling length-based rewards based on group accuracy. This method aims to shorten inference length while maintaining accuracy, without adding extra stages or data. Through multiple experiments on logical and mathematical reasoning tasks, the results show a significant reduction in response length while achieving the same or better performance.

**Strengths:**

1. The problem setting and motivation are clear. Aiming on the reasoning efficiency of LRMs and pointing out the defects of existing length optimization methods.
2. The three reward designs precisely fixe two issues in Kimi's approach. The logic is clear and the approach is highly targeted.
3. The experiments are comprehensive, covering both logic and mathematics, as well as based on multiple baselines. The results show significant length reduction,

**Weaknesses:**

1. The paper claims to have fixed the problem of reduced diversity in the Kimi method and used several methods to measure it, but it lacks the most commonly used methods for measuring diversity and exploration in RL: the entropy of the policy model during training and changes in Pass@K on validation sets.
2. Some test datasets have a small sample size (e.g., AIME with only 30 samples), leading to significant errors in single-run evaluations. However, the experimental results (Table 2) do not report averages across multiple runs, which raises concerns about the validity of the results.
3. The three proposed reward designs are more like empirical engineering optimizations, lacking theoretical derivation. As noted in the paper’s limitations, the method introduces multiple hyperparameters. These require re-tuning whenever the initial policy model is switched or training data is replaced. However, the paper fails to provide clear guidelines for hyperparameter tuning.

**Questions:**

1. As mentioned in weakness 1, can you provide the changes of policy entropy as well as Pass@K to prove the training diversity and exploration are better than baselines?
2. As mentioned in weakness 2, can you provide the average Acc. of multiple evaluations for AIME and AMC? 16 or 32 averages are common settings. For other datasets with larger data volumes, 4-times averaging can be used to eliminate the randomness caused by dynamic sampling of the language model.
3. Can you provide heuristic rules for hyperparameters selection?
4. Can you provide a theoretical analysis and derivation of these reward designs for policy optimization?

---

### Official Review · Reviewer_rc21 · 2025-10-28

**Soundness:** 2
**Presentation:** 3
**Contribution:** 3
**Rating:** 2
**Confidence:** 4

**Summary:**

Short-RL targets the “over-thinking” problem in on-policy RL training of long-reasoning models by embedding three length-control mechanisms directly into the reward function instead of adding extra training stages. It (i) conditions length penalties only on correct samples, (ii) introduces a neutral-length zone that exempts reasonable outputs, and (iii) gates the penalty with a running accuracy threshold to avoid suppressing emergent reasoning. On Logic-RL and three math pipelines the method cuts average step-wise response length by 33–40 % while simultaneously raising final accuracy 1–3 pp over standard RL and outperforming recent length-penalty baselines.

**Strengths:**

* The article analyzes the problems existing in the efficient reasoning training method based on length penalty and proposes some solutions.
* The writing of the article is good, and the overall structure is relatively clear.

**Weaknesses:**

* The paper provides too little introduction to its own method, which affects the novelty and originality of the paper. The paper is more like an experimental report than an academic paper.
* short-RL has no obvious advantages compared with the extreme method.
* The experiment was only conducted on a single model, and the generalizability claimed in the article lacks proof.

**Questions:**

* Reduce the proportion of analysis and increase the proportion of reward design and highlight the differences from related work..
* The paper mentions that the reward design in paper is broadly applicable to other length-based reward functions, adding experiments in this regard will help enhance credibility.
* Conduct more experiments, such as testing whether short-RL can be effective on LLMs from different families and at different stages.

---

### Official Review · Reviewer_1utQ · 2025-11-02

**Soundness:** 2
**Presentation:** 2
**Contribution:** 2
**Rating:** 4
**Confidence:** 4

**Summary:**

This paper proposes several new RL reward design choices for mitigating the _overthinking_ issue in Large Reaoning Models (LRMs). Specifically, it draws inspiration from previous length-based reward designs and notices their drawbacks. They propose to (i) apply the length reward only to correct responses and (ii) introduce a length tolerance parameter to avoid penalizing responses slightly longer than the shortest one (within a rollout group). Noticing the instability of the RL training process, the paper further (iii) incorporates an accuracy threshold to remove the length reward component when the batch-wise accuracy is below a threshold.

**Strengths:**

- The method is overall straight-forward and easy to understand. And it does get motivated from a realistic & important issue (overthinking) in today's LRMs.
- The authors conduct experiments on both the logical and the mathematical domains.
- The writing is generally clear and easy to follow.

**Weaknesses:**

- The authors make several (IMO subjective) assumptions, e.g. assuming that applying length reward to incorrect responses can hinder exploration (Sec 3.2.1) and hypothesizing that including length reward at every step causes training instability (Sec 3.2.2). While I understand that the authors want to motivate their novel designs for highlighting some potential issues with existing approaches, note that these practices (i.e. having length reward for every step & negative examples) are prevalent in many recent works. So some solid empirical results (other than simply using Kimi's failure case as evidence) are needed to justify the assumption.
- For evaluation, why are the "step-wise average response length during training" and "average token length at the last step" -- two training metrics -- used? The average rollout length during test-time should serve as a better and more consistent (with the accuracy metric) choice.
- The training details are somewhat lacking (e.g. even in Appendix only 3 lines are given for the math training). Importantly, what are the training datasets?
- There's also a lack of qualitative assessment of the trained model (which is important for efficient reasoning works): what parts of the reasoning trace get reduced the most? How does the length reduction behave for easier versus more difficult tasks?
- It seems that (from Figure 5) the two hyerparameter choices do have a non-trivial influence on response lengths and accuracies. Are these ablations performed on the logical task or the math task? Are the optimal hyer-parameters very different across different reasoning tasks?

**Questions:**

All of my primary questions are included within the `weaknesses` section. Other than that, please make sure to correct many messy citations (the authors should use `\citep` for lots of citations) in the current text.

---

### Official Review · Reviewer_Gb2D · 2025-11-02

**Soundness:** 2
**Presentation:** 2
**Contribution:** 2
**Rating:** 2
**Confidence:** 4

**Summary:**

The paper dives deep into the reward design in RL for controlling the response length of large reasoning models. Starting from the reward design of K1.5, it identifies three obstacles that hinder the effectiveness of length reward, and adopts solutions for the three problems, respectively. The experimental results show that the effectiveness of the proposed solutions, overall, are called short-RL.

**Strengths:**

1. The motivation for the proposed modification to the length reward is clearly presented.
2. The effectiveness of three designs in short-RL are verified.

**Weaknesses:**

1. The contribution seems to be marginal, which makes it not very suitable for this venue. Essentially, the reward designs (2) and (3) are proposed in this paper; the first design is similar to those in previous works, as discussed in lines 214-215.

2. Unclear reason behind the specific design choice. Although the motivation and problems are clearly discussed for each reward design, when the reward is designed to be the current one, and why other potential choices are not adopted need further discussion in this paper. For example, the use of $acc >= acc_{max} - \tau_{acc}$ needs more explanation to be understood.

3. The measurement of token length is indirect. The accuracy is reported based on a checkpoint during training, but the token length is reported based on step-wise average response length during training and average response length at the final step. If the model is picked with early stop, i.e., it is not the final step, the response length at the final step is meaningless and does not correlate with the model to be tested. Furthermore, we should take care of how many tokens are consumed when producing the performance for each specific benchmark. Therefore, it is more natural to directly compute the length of responses to the questions for each test benchmark. A clearer comparison should be like Table 1 of S-GRPO[1], which reports the accuracy and the average response token length for each specific benchmark.

4. Small issues in writing:
    1. Inconsistent use of symbols: In Eq. 1, the reward of length is noted as $S(y)$, while in other places it is also noted as $reward_{len}(i)$.
    2. In lines 305-306, AIME2024invitational mathematics and AMC23-AI-MO seems to be a typo, it seems to be misuse of $\cite$ and $\citep$.
    3. No conclusion in the paper, though there is still space left on page 9.

[1] S-GRPO: Early Exit via Reinforcement Learning in Reasoning Models

**Questions:**

1. The reviewer cannot understand Reward Design III: uses the condition $acc >= acc_{max} - \tau_{acc}$. When $acc_{max} <  \tau_{acc}$, what will happen？When $acc_{max}$ grows with training, the threshold for adopting length regularization also seems to increase. Is it expected?
2. Why not directly use $\tau_{acc}$ or make the threshold for adopting length regularization decrease when $acc_{max}$ increase?

---

### Note · Authors · 2026-01-23

I have read and agree with the venue's withdrawal policy on behalf of myself and my co-authors.